# LAYER-WISE DYNAMIC RANK FOR COMPRESSING LARGE LANGUAGE MODELS

## ABSTRACT

Large language models (LLMs) have rapidly scaled in size, bringing severe memory and computational challenges that hinder their deployment. Singular Value Decomposition (SVD)-based compression has emerged as an appealing post-training compression technique for LLMs, yet most existing methods apply a uniform compression ratio across all layers, implicitly assuming homogeneous information included in various layers. This overlooks the substantial intra-layer heterogeneity observed in LLMs, where middle layers tend to encode richer information while early and late layers are more redundant. In this work, we revisit the existing SVD-based compression method and propose D-Rank, a framework with layer-wise balanced **D**ynamic **Rank** allocation for LLMs compression. We first introduce effective rank as a principled metric to measure the information density of weight matrices, and then allocate ranks via a Lagrange multiplier-based optimization scheme to adaptively assign more capacity to groups with higher information density under a fixed compression ratio. Moreover, we rebalance the allocated ranks across attention layers to account for their varying importance and extend D-Rank to latest LLMs with grouped-query attention. Extensive experiments on various LLMs with different scales across multiple compression ratios demonstrate that D-Rank consistently outperforms SVD-LLM, ASVD, and Basis Sharing, achieving more than 15 lower perplexity with LLaMA-3-8B model on C4 datasets at 20% compression ratio and up to 5% higher zero-shot reasoning accuracy with LLaMA-7B model at 40% compression ratio while achieving even higher throughput.

## 1 INTRODUCTION

As large language models (LLMs) expand in both scale and deployment, their associated computational and environmental costs continue to escalate (Fernandez et al., 2025). For example, a 30B-parameter model (e.g., LLaMA-30B) requires about 66GB for FP16 weights, which exceeds the capacity of a single GPU and forcing the adoption of model parallelism across multiple GPUs (Touvron et al., 2023a). And a 176 billion parameter model BLOOM running on Google Cloud received 230,768 queries over 18 days, using an average of 40.32 kWh per day (roughly equivalent to 1,110 smartphone charges), demonstrating the substantial energy requirements for model inference at scale (Luccioni et al., 2024; 2023). To mitigate these costs, model compression techniques (e.g., pruning (Sun et al., 2024; Zhang et al., 2024; Ling et al., 2024; Frantar & Alistarh, 2023; Petri et al., 2023) , quantization (Sun et al., 2023; Zhao et al., 2024; Xiao et al., 2023; Lin et al., 2024; Ashkboos et al., 2024b), knowledge distillation (Gu et al., 2024; Magister et al., 2023; Jiang et al., 2023b; Qiu et al., 2024)) that reduce computational and storage demands while preserving model accuracy have been extensively employed to facilitate more efficient LLM deployment. Although effective, these approaches typically require time-consuming retraining process and specialized hardware configurations (e.g., 2:4 semi-structured deployment for GPU-based pruning), creating practical deployment bottlenecks (Li et al., 2023; Ma et al., 2023).

As an effective solution to these limitations, compression techniques such as low-rank adaptation with Singular Value Decomposition (SVD) (Meng et al., 2024) or QR decomposition (Saha et al., 2024) have been extensively employed in LLM deployment (Bałazy et al., 2025). In SVD-based low-rank adaptation, each weight matrix is approximated through decomposition into three matrices with much smaller dimensions (Yuan et al., 2025; Wang et al., 2025b). After the decomposition,

matrix multiplications are performed on the lower-dimensional factors rather than the original full matrix, resulting in substantial parameter reduction and improved storage and computation efficiency while preserving model performance comparable to the original full-rank model. Typically, a weight matrix $W \in \mathbb{R}^{m \times n}$ can be approximated as $W \approx U_k \Sigma_k V_k^\top$, where $k < \min(m, n)$ denotes the retained rank. A larger compression ratio will lead to a smaller $k$.

Despite the benefits and popularity of SVD-based low-rank adaptation in practical LLM deployment scenarios, there has been limited research on how to define an effective metric to quantify the information content in weight matrices, and subsequently attain dynamic ranks by leveraging the differences in information content between intra-layer attention matrices and cross-layer matrices, which therefore can maximize information preservation under a given overall compression ratio. In this work, we observe and identify several bottlenecks that hinder the efficiency of current SVD-based low-rank adaptation approaches: 1) limited effort has been devoted to designing effective metrics for measuring weight information content to determine optimal retained ranks $k$ for each weight, leading to suboptimal compression performance; 2) existing approaches maintain uniform compression ratios across weight matrix types, failing to account for the substantial differences in their information density and inherent complexity; 3) in the latest LLMs with grouped-query attention, compression techniques such as grouping weight matrices across layers for joint compression may become ineffective due to substantial reduction in the column dimension of the $W_K$ and $W_V$ weight matrices compared with Multi-Head-Attention-based architectures (MHA), yet there is a lack of explanation for the underlying reasons as well as corresponding optimization strategies.

To address the bottleneck , we develop the layer-wise dynamic rank for SVD-based LLMs compression. Specifically, we propose a metric, effective rank, for measuring information density of weight matrices. Subsequently, the effective rank will be employed to guide us in dynamically adjusting the retained rank for different types of weight matrices across different layers. To further improve the compression performance, we reallocate the preserved ranks across matrix types for attention layers by transferring part of the budget from matrices with lower information density to ones with higher information density while the same the overall target compression ratio. The main contributions of this work can be summarized as follows:

- We propose **D-Rank**, a layer-wise dynamic rank allocation approach for compressing LLMs. This approach enables us to preserve more information in large language models under the same compression budget, thereby achieving superior compression performance.

- We introduce a novel metric, effective rank, to quantify the information density of each grouped layer in LLMs. Moreover, we develop a Lagrangian multiplier-based framework that dynamically allocates ranks across grouped layers according to their effective rank, aiming to improve the information preserved in the models.

- Through effective rank analysis, we discover that the effective rank distribution among the attentions matrices is highly unbalanced: $W^Q, W^K$ have lower effective rank (less information) than $W^V$ matrix. To address this issue, we propose a reallocation strategy that transfers part of the preserved rank budget from $W^Q, W^K$ to $W^V$.

- Moreover, we analyze the reason why the performance of latest models (e.g., LLaMA-3) with grouped-query attention degrades in the state-of-the-art works, and we further demonstrate the effectiveness of D-Rank on the models.

Extensive experiments on the LLaMA, LLaMA-2, LLaMA-3, and Mistral families show that D-Rank consistently outperforms baselines, achieving more than 15 lower perplexity on LLaMA-3-8B model with 20% compression ratio on C4 datasets, and up to 5% higher accuracy on zero-shot reasoning tasks with LLaMA-7B model at 40% compression ratio, while it has even higher token throughput compared to baselines.

## 2 MOTIVATION AND RESEARCH QUESTIONS

Recent research (Hu et al., 2025; Gao et al., 2024; Razzhigaev et al., 2023; Wei et al., 2024) show that the **information content** of weight matrices varies significantly across layers. For example, studies show that with respect to the input activations $X$, early and late layers of LLMs exhibit lower information density, while middle layers contain substantially richer information, forming a characteristic U-shaped distribution across depth (Razzhigaev et al., 2023; Hu et al., 2025). Although

layer-wise information differences in LLMs have been discussed in other applications, in model compression, few works have investigated how to design metrics which can effectively quantify such differences across different weight matrices, and how to leverage these metrics to develop effective allocation strategies for layer-wise rank allocation. Then we naturally raise the following question:

> **Question 1**
>
> *How does the **information content** of weights vary across layers? **What metric** should we use to quantify this information content, and **how** can it guide adaptive rank allocation for compression?*

Prior work demonstrates that attention layers in Transformer-based models exhibit substantial redundancy and notable inter-layer heterogeneity (Voita et al., 2019). Additionally, different attention matrices in Transformer-based models show extremely unbalanced importance during fine-tuning, indicating that the effective parameter space varies significantly across different matrices in attention layers (Yao et al., 2024). Recently, several parameter efficient fine-tuing (PEFT) works (Zhang et al., 2023; Liu et al., 2024b) tend to allocate ranks or parameter budgets adaptively across layers and individual attention matrices across attention layers, consistently outperforming uniform rank allocation, and empirically demonstrating that different matrices possess varying levels of importance. However, most existing SVD-based model compression works apply identical compression ratios (or ranks) to all attention layer weight matrices, with limited exploration of inter-layer heterogeneity for adaptive compression ratio distribution. This motivates the following question:

> **Question 2**
>
> *Do different weight matrices in attention layers, especially $W^Q$, $W^K$, and $W^V$, contain different levels of information, and should non-uniform ranks be allocated for attention layer compression?*

## 3 METHODOLOGY

### 3.1 NOTATION AND PRELIMINARY

Assume an LLM model with $N$ layers and $G$ groups, and for each group there are $n$ layers, the weight of $i$-th layer inside of a group is denoted as $W^{(i)} \in \mathbb{R}^{d_1 \times d_2}$. We can first concatenate matrices within the same group horizontally: $W = \begin{bmatrix} W^{(1)} & W^{(2)} & \dots & W^{(n)} \end{bmatrix} \in \mathbb{R}^{d_1 \times (nd_2)}$. We then perform SVD: $W = U\Sigma V^\top$. After truncating to the top $k$ singular values, we obtain: $W \approx W_k = U_k \Sigma_k V_k^\top$, where $U_k \in \mathbb{R}^{d_1 \times k}, \Sigma \in \mathbb{R}^{k \times k}, V_k^\top \in \mathbb{R}^{k \times nd_2}$. We define $B = U_k \Sigma_k \in \mathbb{R}^{d_1 \times k}$ as the shared basis matrix and split $V_k^\top$ into blocks $C^{(i)} \in \mathbb{R}^{k \times d_2}$ as the layer-specific coefficient matrices: $W^{(i)} \approx B\,C^{(i)}$. That is, each column of $W^{(i)}$ is expressed as a linear combination of $k$ shared basis vectors: $W_{:,j}^{(i)} \approx \sum_{m=1}^{k} B_{:,m} C_{m,j}^{(i)}$.

However, directly applying SVD on the weight matrix without considering the effects of the calibration data on activation $X$ is impractical since this might lead to significant compression loss and potentially affect the performance of the LLM after compression (Wang et al., 2025a). Therefore, several works (Yuan et al., 2025; Wang et al., 2025b) propose that we can incorporate the input activation statistical information $\mathcal{S}$ for SVD calculation, which can be expressed as $SS^\top = \text{cholesky}(X^\top X)$ and $W = \mathcal{S}^{-1}(\mathcal{S}W)$. Following these works (Wang et al., 2025a;b), we apply SVD to the scaled matrix $\mathcal{S}W$ instead of $W$: $\mathcal{S}W \approx U_k'\Sigma_k'V_k'^\top$, and we can reconstruct $W$ as $W \approx \mathcal{S}^{-1}U_k'\Sigma_k'V_k'^\top = B''C'$, where $B'' = \mathcal{S}^{-1}U_k'\Sigma_k'$ is the shared basis matrix and $C'$ are the coefficient matrices. Notably, when $n = 1$ only, the procedure is the standard SVD-LLM approach.

### 3.2 LAYER-WISE DYNAMIC RANK SELECTION VIA EFFECTIVE RANK-BASED INFORMATION DENSITY CALCULATION

#### 3.2.1 EFFECTIVE RANK FORMULATION

Considering the $g$-th group of matrices is denoted as $W_g \in \mathbb{R}^{d_1 \times nd_2}$, the effective rank of $W_g$ is calculated based on the spectral entropy of the scaled matrix $\mathcal{S}_g W_g$. We first calculate the $i$-th squared singular values of $\mathcal{S}_g W_g$ as $\lambda_g^i = (\sigma_g^i)^2$ ($0 \le i < d_1$), , which represent the energy along the

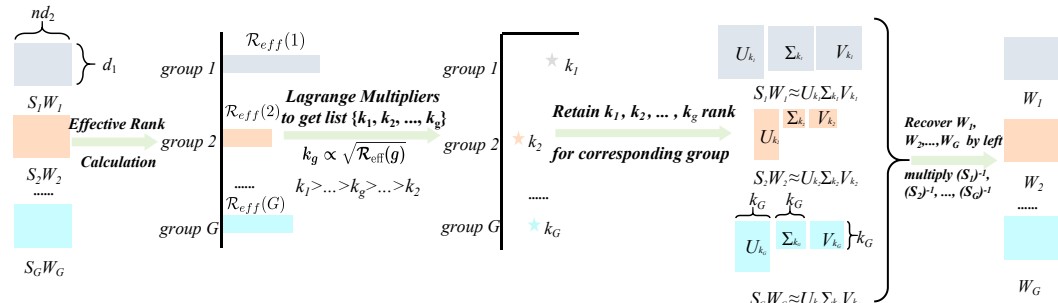

Figure 1: The overall pipeline of our proposed D-Rank

$i$-th principal component. These squared singular values are then normalized to form a probability distribution $\mathcal{P}$, and the $i$-th element of the distribution is defined as:

$$p_g^i = \frac{\lambda_g^i}{\sum_j \lambda_g^i} \tag{1}$$

We further define the effective rank $\mathcal{R}_{\text{eff}}$ as to evaluate the sensitivity of group $g$ using the exponential of the Shannon entropy of the distribution, which is used to measure the number of significant singular values of the scaled matrix $\mathcal{S}_g W_g$. We formulate the effective rank as:

$$\mathcal{R}_{\text{eff}}(g) = \exp\left(-\sum_i p_g^i \log p_g^i\right) \tag{2}$$

The formulation considers the overall singular value distribution of each scaled matrix $\mathcal{S}_g W_g$, which can be regarded as the information density of it. We use the effective rank $\mathcal{R}_{\text{eff}}(g)$ to represent the minimum number of singular values required to effectively represent the uncompressed scaled matrix $\mathcal{S}_g W_g$. A lower effective rank indicates higher redundancy, while a higher effective rank suggests higher information density of the group.

### 3.2.2 RANK ALLOCATION VIA LAGRANGE MULTIPLIERS

**Motivation.** Table 1 shows that the effective rank varies substantially across different layer groups, indicating the non-uniform information density over depth. In particular, the middle layers generally have higher effective ranks than the earlier and the later layers, which is consistent with existing studies showing the U-shaped information distribution across depth in Transformer-based models(Razzhigaev et al., 2023; Hu et al., 2025). Such variability implies that applying a single, uniform compression ratio to all groups may be suboptimal, as it ignores depth-wise information density difference. For better performance with a fixed overall compression ratio, guided by effective rank, we allocate each group's retained rank $k_g$ based on our proposed **rank allocation via Lagrangian multipliers**.

Table 1: Effective rank of grouped matrices for $V, K, Q$ in LLaMA-7B on Wikitext-2 (two layers as a group)

| Group Index | V | K | Q |
|---|---|---|---|
| 1 | 118 | 6 | 7 |
| 3 | 592 | 8 | 12 |
| 7 | 778 | 12 | 33 |
| 10 | 1026 | 15 | 24 |
| 12 | 973 | 12 | 25 |
| 14 | 1148 | 11 | 29 |
| 16 | 846 | 10 | 23 |

This reallocation maximizes the information preserved in the model after compression under a fixed target compression ratio. And the group with a higher effective rank $\mathcal{R}_{\text{eff}}(g)$ will be assigned a larger budget proportionally.

Suppose a LLM model with $N$ layers and $G$ groups, and for each group there are $n$ layers. For the $i$-th group, we have $\mathcal{R}_{\text{eff}}(g)$ as the effective rank to quantify the information density of the group. We denote $\omega$ as the parameter cost per rank to represent the number of parameters required to increase the rank of the group by one. For a shared basis, this is calculated as $\omega = d_1 + nd_2$, where $n$ is the number of layers in the group. We use $k_g$ to denote the number of ranks to be allocated to group $g$. We further define a total reallocation error as $\ell_{\text{total}}$, which is used as the penalty of the distribution inconsistency of the allocated rank with the effective rank accumulated across all groups, with the assumption that the error is inversely proportional to the allocated rank and proportional to the effective rank. Suppose the total number of parameters of all groups is $\mathcal{T}$ and the target compression ratio is $\theta$, we

denote $\mathcal{T}_{\text{budget}} = \mathcal{T}(1 - \theta) = \sum_{g=1} k_g \omega$ as the total number of parameter in the compressed module. We then formulate the optimization problem as:

$$\underset{k_1, k_2, \ldots}{\text{minimize}} \quad \ell_{\text{total}} = \sum_{g=1} \frac{\mathcal{R}_{\text{eff}}(g)}{k_g}$$

$$\text{subject to} \quad \sum_{g=1} k_g \omega = \mathcal{T}_{\text{budget}} \tag{3}$$

Using Lagrange multipliers, we can solve the constrained optimization problem with the following Lagrange function:

$$\mathcal{F}(\{k_g\}, \lambda) = \sum_{g=1} \frac{\mathcal{R}_{\text{eff}}(g)}{k_g} + \lambda \left( \sum_{g=1} k_g \omega - \mathcal{T}_{\text{budget}} \right) \tag{4}$$

$\lambda$ is the Lagrangian multiplier. Taking the derivative of $\mathcal{F}$ with respect to each $k_g$ and setting it to zero:

$$\frac{\partial \mathcal{F}}{\partial k_g} = -\frac{\mathcal{R}_{\text{eff}}(g)}{k_g^2} + \lambda \omega = 0 \tag{5}$$

The solution reveals that the optimal rank $k_g$ for each group should be determined according to the following proportionality:

$$k_g \propto \sqrt{\frac{\mathcal{R}_{\text{eff}}(g)}{\omega}} \tag{6}$$

We can see that the optimal rank is proportional to the square root of the group's $\mathcal{R}_{\text{eff}}(g)$ and inversely proportional to the square root of its parameter cost (more expensive groups get fewer ranks). Applying the budget constraint, we have

$$k_g = \frac{\mathcal{T}_{\text{budget}}}{\sum_{j=1} \sqrt{\mathcal{R}_{\text{eff}}(j)\,\omega}} \cdot \frac{\sqrt{\mathcal{R}_{\text{eff}}(g)}}{\sqrt{\omega}}, \tag{7}$$

Afterwards, we obtain a list $\mathcal{L}$ that records the retained rank required for each group of such weight matrices $[k_1, k_2, ..., k_G]$. Detailed allocation strategy is shown in Appendix B.3. The layer-wise dynamic rank selection pipeline is illustrated in Figure 1. First, for each group, weight matrices across $n$ layers are concatenated horizontally and multiplied by $\mathcal{S}$ to form scaled matrix $\mathcal{S}_g W_g$ ($\mathcal{S}$ is calculated by $\mathcal{S}\mathcal{S}^\top = \text{cholesky}(X^\top X)$ from activations $X$ and $g$ is the index of the group), then we calculate the effective rank of each group of scaled matrix. After we get the rank $\{k_1, k_2, ..., k_G\}$ for each grouped matrix with Lagrange Multiplier, we will use them as the singular values to perform the SVD compression for every scaled weight matrix.

### 3.3 BALANCING DYNAMIC RANK ACROSS ATTENTION LAYERS

**Motivation.** We group every two layers of LLaMA-7B and estimate the effective ranks of $W^Q, W^K, W^V$, as shown in Figure 2. We observe that $W^V$ consistently exhibits much larger $\mathcal{R}_{\text{eff}}$ (which is often $> 1000$) than $W^Q, W^K$ indicating that the information density is unevenly distributed across the attention layers. This observation motivates us to consider two key questions: *do different weight matrices show substantial disparities in information density, and can such disparities inform how we adjust compression ratios across matrices?*

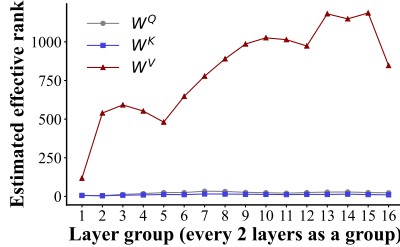

Figure 2: Effective ranks of grouped $W^Q, W^K, W^V$ matrices for LLaMA-7B model on Wikitext-2 (two layers as a group)

As discussed in the previous section, the value of $\mathcal{R}_{\text{eff}}$ represents the minimum number of *top-k* singular values to effectively represent the matrix. Therefore, assigning the number of retained singular values $k$ solely based on the effective ranks of each type of matrix according to the Lagrangian method would be unfair to the $W^V$ matrices.

To address this, after computing the number of retained singular values $k$ for each group of the $W^Q$, $W^K$, and $W^V$ matrices using the Lagrangian allocation method, we reallocate part of the $k$ originally assigned to the $W^Q$ and $W^K$ matrices to the $W^V$ matrices. Suppose for a LLaMA-7B

models (32 layers in total) has 4 groups, then each group has 8 layers. Based on our proposed rank allocation via Lagrange multipliers, we can obtain the list $\mathcal{L}$ of reallocated rank $W^k$ for each group of $W^Q, W^K, W^V$ as follows:

$$\mathcal{L}^Q = [k_1^Q, k_2^Q, k_3^Q, k_4^Q], \quad \mathcal{L}^K = [k_1^K, k_2^K, k_3^K, k_4^K], \quad \mathcal{L}^V = [k_1^V, k_2^V, k_3^V, k_4^V] \tag{8}$$

We then define **an adjustment ratio** $\beta$ and extract a portion of the rank proportional to $\beta$ from the $\mathcal{L}^Q$ and $\mathcal{L}^K$, respectively. Then we sum up the extracted rank, and redistribute the accumulated rank evenly across the elements of $\mathcal{L}^V$:

$$\mathcal{L}_{\text{final-}k}^Q = (1 - \beta)[k_1^Q, k_2^Q, k_3^Q, k_4^Q], \tag{9}$$

$$\mathcal{L}_{\text{final-}k}^K = (1 - \beta)[k_1^K, k_2^K, k_3^K, k_4^K], \tag{10}$$

$$t = \frac{\beta}{4} \left( \sum_{i=1} k_i^Q + \sum_{i=1} k_i^K \right), \tag{11}$$

$$\mathcal{L}_{\text{final-}k}^V = [k_1^V + t, k_2^V + t, k_3^V + t, k_4^V + t]. \tag{12}$$

Then, we obtain the final adjusted numbers of retained singular values, for the $W^Q$, $W^K$, and $W^V$ matrices in the attention layers. Overall, since the $W^V$ matrices generally exhibit higher effective ranks than the $W^Q$ and $W^K$ matrices, this adjustment allows the $W^V$ matrices, which require more information capacity, to retain higher singular values. The parameter $\beta$ serves as a tunable hyperparameter, and we will provide a detailed analysis of its impact in the experimental section.

### 3.4 Dynamic Rank Allocation for Models with Grouped-Query Attention

We observe that when the number of layers in each group increases, there is a trend that the performance will decrease (i.e., ppl increases) on LLaMA-3, as shown in Table 2. We analyze the reasons as follows: 1) On LLaMA-3-8B, the $W^K, W^V$ projection matrices have dimensions of $4096 \times 1024$. When such matrices are horizontally concatenated within a group, the dimension expands severely and the matrix rank could be even larger than the rank of any indi-

Table 2: Evaluation of PPL($\downarrow$) of LLaMA-3-8B on Wikitext-2 under 20% and 30% compression ratio

| Method | Grouped layers | 20% | 30% |
|---|---|---|---|
| SVD-LLM | 1 | 15.45 | 30.59 |
| Basis Sharing | 2 | 14.70 | 31.87 |
| | 3 | 20.28 | 55.29 |
| | 4 | 22.57 | 66.94 |
| | 5 | 17.09 | 44.09 |

vidual matrix. Under a fixed compression ratio, the resulting SVD truncation will lead to a larger reconstruction error for the concatenated matrix than for compressing the original per-layer matrices separately. 2) Since the $W^K, W^V$ projections in LLaMA-3 are architecturally slimmed to reduce KV-cache memory compared to LLaMA and LLaMA-2 (Touvron et al., 2023a), grouping $n > 1$ layers for joint SVD results in fewer retained ranks per matrix under a fixed global compression ratio, leading to more aggressive compression of individual matrices. For example, at a 20% compression ratio, $n = 1$ retains $k = 655$ ranks per group, while $n = 2$ yields $k = 1092$ group ranks (around 546 per matrix) (Wang et al., 2025a), demonstrating the more aggressive per-matrix compression.

To address the issue, for models with grouped-query attentions, such as LLaMA-3, we set the group size as $n = 1$, and we use our proposed compression scheme that (i) dynamically adjusts the retained rank $k$ of each layer according to its effective rank; (ii) reallocates a portion of the $k$ budget from the $W^Q$ and $W^K$ matrices to the $W^V$ matrices. Our experimental results demonstrate that the proposed method remains effective on LLaMA-3 architecture models.

## 4 Experiments

### 4.1 Experimental Setting

**Datasets.** For language modeling, we use three datasets in our experiments: PTB, WikiText2, and C4 ((Marcus et al., 1993); (Merity et al., 2017); (Raffel et al., 2020)). To evaluate the model's reasoning ability, we employ seven reasoning datasets: MathQA, PIQA, ARC-e, ARC-c, HellaSwag, WinoGrande, and OpenbookQA ((Amini et al., 2019); (Bisk et al., 2019); (Clark et al., 2018); (Zellers et al., 2019); (Sakaguchi et al., 2021); (Banerjee et al., 2019)). The LM-Evaluation-Harness framework has been applied to test every reasoning task through a zero-shot setting (Sutawika et al., 2024).

Table 3: Comparison of PPL(↓) and Zero-shot(↑) performance of LLaMA-7B with baselines. The $\mathcal{S}$ of all tasks is obtained with the dataset WikiText-2 and $n = 2$

| RATIO | Method | WikiText-2↓ | PTB↓ | C4↓ | Openb.↑ | ARC_e↑ | WinoG.↑ | HellaS.↑ | ARC_c↑ | PIQA↑ | MathQA↑ | Average*↑ |
|---|---|---|---|---|---|---|---|---|---|---|---|---|
| 0% | Original | 5.68 | 8.35 | 7.34 | 0.28 | 0.67 | 0.67 | 0.56 | 0.38 | 0.78 | 0.27 | 0.47 |
| 20% | SVD | 20061 | 20306 | 18800 | 0.14 | 0.27 | 0.51 | 0.26 | 0.21 | 0.53 | 0.21 | 0.31 |
| | FWSVD | 1727 | 2152 | 1511 | 0.15 | 0.31 | 0.50 | 0.26 | 0.23 | 0.56 | 0.21 | 0.32 |
| | ASVD | 11.14 | 16.55 | 15.93 | 0.25 | 0.53 | 0.64 | 0.41 | 0.27 | 0.68 | 0.24 | 0.43 |
| | SVD-LLM | 7.94 | 18.05 | 15.93 | 0.22 | 0.58 | 0.63 | 0.43 | 0.29 | 0.69 | 0.24 | 0.44 |
| | Basis Sharing | 7.74 | 17.35 | 15.03 | 0.28 | 0.66 | 0.66 | 0.46 | **0.36** | 0.71 | **0.25** | 0.48 |
| | **D-Rank (Ours)** | **7.45** | **15.99** | **13.73** | **0.29** | **0.69** | 0.66 | **0.47** | 0.36 | **0.72** | 0.25 | **0.49** |
| 30% | SVD | 13103 | 17210 | 20871 | 0.13 | 0.26 | 0.51 | 0.26 | 0.21 | 0.54 | 0.22 | 0.30 |
| | FWSVD | 20127 | 11058 | 7240 | 0.17 | 0.26 | 0.49 | 0.22 | 0.22 | 0.51 | 0.19 | 0.30 |
| | ASVD | 51 | 70 | 41 | 0.18 | 0.43 | 0.53 | 0.37 | 0.25 | 0.65 | 0.21 | 0.38 |
| | SVD-LLM | 9.56 | 29.44 | 25.11 | 0.20 | 0.48 | 0.59 | 0.40 | 0.26 | 0.65 | 0.22 | 0.40 |
| | Basis Sharing | 9.25 | 29.12 | 22.46 | 0.27 | 0.63 | 0.63 | 0.40 | 0.30 | 0.68 | 0.24 | 0.45 |
| | **D-Rank (Ours)** | **8.97** | **26.40** | **20.44** | 0.28 | **0.65** | **0.64** | **0.42** | **0.32** | **0.69** | **0.25** | **0.46** |
| 40% | SVD | 52489 | 59977 | 47774 | 0.15 | 0.26 | 0.52 | 0.26 | 0.22 | 0.53 | 0.20 | 0.30 |
| | FWSVD | 18156 | 20990 | 12847 | 0.16 | 0.26 | 0.51 | 0.26 | 0.22 | 0.53 | 0.21 | 0.30 |
| | ASVD | 1407 | 3292 | 1109 | 0.13 | 0.28 | 0.48 | 0.26 | 0.22 | 0.55 | 0.19 | 0.30 |
| | SVD-LLM | 13.11 | 63.75 | 49.83 | 0.19 | 0.42 | 0.58 | 0.33 | 0.25 | 0.60 | 0.21 | 0.37 |
| | Basis Sharing | 12.39 | **55.78** | 41.28 | 0.22 | 0.52 | **0.61** | 0.35 | **0.27** | 0.62 | **0.23** | 0.40 |
| | **D-Rank (Ours)** | **11.99** | 56.04 | **37.22** | **0.23** | **0.57** | **0.61** | **0.36** | 0.27 | **0.64** | 0.23 | **0.42** |
| 50% | SVD | 131715 | 87227 | 79815 | 0.16 | 0.26 | 0.50 | 0.26 | 0.23 | 0.52 | 0.19 | 0.30 |
| | FWSVD | 24391 | 28321 | 23104 | 0.12 | 0.26 | 0.50 | 0.26 | 0.23 | 0.53 | 0.20 | 0.30 |
| | ASVD | 15358 | 47690 | 27925 | 0.12 | 0.26 | 0.51 | 0.26 | 0.22 | 0.52 | 0.19 | 0.30 |
| | SVD-LLM | 23.97 | 150.58 | 118.57 | 0.16 | 0.33 | 0.54 | 0.29 | 0.23 | 0.56 | 0.21 | 0.33 |
| | Basis Sharing | 20.00 | 126.35 | 88.44 | 0.18 | 0.42 | 0.57 | 0.31 | 0.23 | **0.58** | **0.22** | 0.36 |
| | **D-Rank (Ours)** | **19.82** | **126.10** | **80.69** | **0.20** | **0.46** | **0.58** | **0.32** | **0.24** | 0.58 | 0.22 | **0.37** |

Table 4: PPL(↓) and Zero-shot(↑) performance on LLaMA-3-8B under 20% compression ratio. The $\mathcal{S}$ of all tasks is obtained with the dataset WikiText-2. For Basis sharing baseline, $n = 5$

| Method | WikiText-2↓ | C4↓ | Openb.↑ | ARC_e↑ | WinoG.↑ | HellaS.↑ | ARC_c↑ | PIQA↑ | MathQA↑ | Average*↑ |
|---|---|---|---|---|---|---|---|---|---|---|
| Original | 6.14 | 9.47 | 0.34 | 0.75 | 0.70 | 0.57 | 0.40 | 0.79 | 0.27 | 0.55 |
| FWSVD | 4782 | 8195 | 0.01 | 0.04 | 0.01 | 0.02 | 0.01 | 0.02 | 0.01 | 0.02 |
| ASVD | 17.55 | 77.25 | 0.20 | 0.59 | 0.61 | 0.41 | 0.28 | 0.68 | 0.24 | 0.43 |
| SVD-LLM | 15.45 | 78.01 | 0.24 | 0.63 | 0.62 | 0.40 | 0.30 | 0.68 | 0.27 | 0.45 |
| Basis Sharing | 17.09 | 60.08 | 0.25 | 0.65 | 0.66 | 0.40 | 0.31 | 0.69 | 0.26 | 0.46 |
| **D-Rank (Ours)** | **13.68** | **44.87** | **0.27** | **0.68** | **0.67** | **0.43** | **0.33** | **0.71** | **0.28** | **0.48** |

**Models.** We conduct comprehensive evaluations of D-Rank across multiple LLMs, including the LLaMA family (LLaMA-7B, LLaMA-13B, LLaMA-30B, LLaMA-2-7B, LLaMA-3-8B)((Touvron et al., 2023a); (Touvron et al., 2023b); (Dubey et al., 2024)) and Mistral-7B((Jiang et al., 2023a)).

**Baselines.** We contrast comparative evaluations with existing methods that utilize SVD-based weight approximation in individual layers without cross-layer parameter sharing. We specifically benchmark against FWSVD (Hsu et al., 2022), ASVD (Yuan et al., 2025), SVD-LLM (Wang et al., 2025b), and Basis Sharing (Wang et al., 2025a).

**Implementation Details and Hyperparameter Settings.** All experiments are conducted on two NVIDIA A100 80GB GPUs. The LLaMA-30B model is implemented in FP16 precision, while all other models utilize FP32 precision. We use FP64 to maintain the computational precision of matrix $\mathcal{S}$. Matrix $\mathcal{S}$ is derived from 256 samples of WikiText-2 with a sequence length of 2048. Note that when the compression ratio is 40% or more, accumulated compression errors lead to significant inter-layer input deviation from original values. We adaptively update the downstream layer weights using the deviated inputs, similar to the method used in SVD-LLM. Following (Wang et al., 2025a), matrices like $W^Q$, $W^K$, $W^V$, $W^{\text{up}}$, and $W^{\text{gate}}$ in MHA-based models are grouped and compressed in our experiments when $n > 1$, while $W^{\text{down}}$ and $W^O$ are not grouped.

## 4.2 MAIN RESULTS

**Performance on generation and reasoning datasets.** On LLaMA-7B with $S$ from Wikitext-2 and group size $n = 2$, D-Rank consistently has a better performance under 20–50% compression compared with baselines as shown in Table 3. Compared with SVD-LLM, we reduce PPL on Wikitext-2, PTB and C4 by 6–32% across ratios. For instance, at 20% compression ratio D-Rank can achieve about 0.5 lower PPL than SVD-LLM and raise average zero-shot accuracy by about 0.11 at 30% compression ratio. Compared with Basis Sharing, our approach attains equal or higher average accuracy at all ratio and typically lower PPL on Wikitext-2 and C4, with a single notable

Table 5: PPL (↓) of different LLMs under 20% compression ratio on WikiText-2

| Method | LLaMA-7B | LLaMA-2-7B | Mistral-7B |
|---|---|---|---|
| SVD | 20061 | 18192 | 159627 |
| FWSVD | 1721 | 2360 | 6357 |
| ASVD | 11.14 | 10.10 | 13.72 |
| SVD-LLM | 7.94 | 8.50 | 10.21 |
| Basis Sharing | 7.74 | 7.70 | 7.57 |
| **D-Rank (Ours)** | **7.45** | **7.51** | **7.41** |

Table 6: PPL (↓) of LLaMA-7B, 13B, 30B under 20% compression ratio on WikiText-2

| Method | 7B | 13B | 30B |
|---|---|---|---|
| SVD | 20061 | 946.31 | 54.11 |
| FWSVD | 1630 | OOM | OOM |
| ASVD | 11.14 | 6.74 | 22.71 |
| SVD-LLM | 7.94 | 6.61 | 5.63 |
| Basis Sharing | 7.75 | 6.47 | 5.47 |
| **D-Rank (Ours)** | **7.45** | **6.30** | **5.33** |

exception on PTB at 30%. D-Rank can even achieve a PPL of 80.69, which is about 8 lower than Basis Sharing. As compression tightens from 20% to 50%, all methods' performance degrades, but ours degrades more gracefully, yielding a stronger accuracy–compression trade-off; PTB is the most compression-sensitive among the language modeling datasets.

We also provide the results of D-Rank on LLaMA-3-8B. As shown in Table 4, D-Rank consistently outperforms all baselines under the 20% compression ratio. Compared with baselines, it achieves notably lower perplexity on WikiText-2 and C4. For example, D-Rank can get the lowest PPL of nearly 45 on C4, which is at least 15 lower than baselines. D-Rank also obtains the best zero-shot accuracies on reasoning tasks such as 71% on PIQA and 67% on WinoGrande. On average, D-Rank delivers the highest overall score of 48%, demonstrating superior performance over baselines.

**Performance on different LLMs.** Table 5 reports results on three representative LLMs under a 20% compression ratio. Conventional SVD-based methods suffer from extremely high perplexities, while SVD-LLM and Basis Sharing provide partial improvements.

In contrast, D-Rank achieves the best overall performance across all models. For instance, on LLaMA-2-7B, D-Rank obtains a PPL of 7.51, outperforming SVD-LLM's PPL of 8.5 and Basis Sharing's PPL of 7.57. Similarly, on Mistal-7B we reach the PPL of 7.41, which is lower than all baselines. These results highlight the robustness of D-Rank across different LLMs.

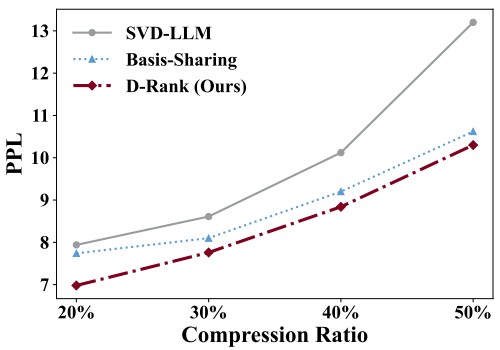

Figure 3: LoRA fine-tuning PPL (↓) results of compressed LLaMA- 7B

**Performance on different scales.** Table 6 further evaluates D-Rank on LLaMA models with three scales of 7B, 13B, and 30B. It can be seen that D-Rank consistently achieves the lowest perplexity. On LLaMA-13B, our approach achieves a PPL of 6.30, lower than 6.61 of SVD-LLM and 6.47 of Basis Sharing. On the largest 30B model, D-Rank yields 5.33, which is better than both Basis Sharing's PPL of 5.47 and SVD-LLM's PPL of 5.63. This demonstrates that D-Rank scales effectively to larger models, maintaining superior accuracy under compression.

**Performance under LoRA fine-tuning.** D-Rank can combine with LoRA fine-tuning to recover performance. Our LoRA fine-tuning settings include $lora\_r = 8$, $lora\_alpha = 32$, and $learning\_rate = 1e-4$, and we use default settings for all other hyperparameters in the Hugging Face PEFT. Each compressed model is fine tuned with WikiText-2 training dataset for two epochs. Figure 3 illustrates the LoRA fine-tuning perplexity (PPL) results of LLaMA-7B with 20–50% compression using different methods. Across all settings, D-Rank consistently yields lower PPL than both SVD-LLM

Table 7: Evaluation of PPL(↓) of LLaMA-7B at 20% compression ratio using C4 as calibration data. Evaluation is done on C4 and Wikitext-2

| Method | Grouped layers | C4 PPL | Wikitext-2 PPL |
|---|---|---|---|
| SVD-LLM | – | 11.84 | 11.60 |
| Basis Sharing | 2 | 11.53 | 10.90 |
| | 3 | 11.44 | 10.98 |
| | 4 | 11.42 | 11.08 |
| | 5 | 11.31 | 11.16 |
| D-Rank (Ours) | 2 | **11.07** | **9.99** |
| | 3 | **10.88** | **10.00** |
| | 4 | **10.78** | **9.78** |
| | 5 | **10.71** | **9.89** |

and Basis-Sharing. The advantage is already evident at 20% compression, and the gap steadily widens as the compression ratio increases. For instance, when compression reaches 50%, our approach reduces PPL by more than 2 compared to SVD-LLM, highlighting its stronger robustness under aggressive compression. These results demonstrate that D-Rank maintains a more favorable accuracy and compression trade-off than existing baselines.

Table 8: Evaluation of PPL($\downarrow$) with different $\beta$ in D-Rank for different grouped layers $n$ on LLaMA-7B under compression ratios from 20% to 50%. $\mathcal{S}$ of all tasks is obtained with WikiText-2

| # Compression ratio | | 20% | | | 30% | | | 40% | | | 50% | |
|---|---|---|---|---|---|---|---|---|---|---|---|---|
| # Grouped layers | | 2 | 3 | 4 | 2 | 3 | 4 | 2 | 3 | 4 | 2 | 3 | 4 |
| Basis Sharing | | 7.74 | 7.72 | 7.65 | 9.25 | 9.27 | 9.18 | 12.39 | 12.60 | 12.58 | 19.99 | 20.06 | 20.86 |
| $\beta$ | 0.2 | 7.51 | 7.53 | 7.40 | 9.04 | 9.00 | 8.93 | 12.11 | 12.13 | 12.08 | 20.19 | 19.60 | 19.72 |
| | 0.25 | 7.48 | 7.42 | 7.36 | 8.99 | 8.98 | 8.91 | 12.08 | 12.11 | 12.06 | 20.05 | 19.53 | 19.65 |
| | 0.3 | **7.45** | **7.37** | 7.37 | **8.97** | 8.99 | **8.87** | 12.03 | 12.00 | 12.04 | 19.87 | 19.46 | 19.49 |
| | 0.35 | 7.47 | 7.40 | **7.35** | 9.00 | **8.89** | 8.90 | **11.99** | **11.98** | **12.02** | 19.85 | **19.32** | 19.41 |
| | 0.4 | 7.50 | 7.39 | **7.35** | 9.07 | 8.93 | 9.02 | 12.04 | 12.01 | 12.07 | **19.83** | 19.39 | **19.35** |
| | 0.45 | 7.54 | 7.39 | 7.36 | 9.12 | 8.96 | 9.04 | 12.06 | 12.03 | 12.10 | 19.89 | 19.53 | 19.46 |

**Performance with calibration data from different datasets.** As shown in Table 7, we use C4 as calibration data to get $\mathcal{S}$ to perform compression on LLaMA-7B at a 20% ratio and then evaluate PPL on both C4 and WikiText-2. We observe that while Basis Sharing achieve moderate reductions in PPL compared to SVD-LLM, D-Rank consistently yields the lowest values across different group sizes. For example, when grouping 4 layers, our approach reduces the PPL on C4 from 11.42 of Basis Sharing to 10.78, and on WikiText-2 from 11.08 to 9.78. This demonstrates that D-Rank not only preserves performance on the Wikitext-2 calibration dataset but also transfers better to out-of-distribution evaluation, highlighting its effectiveness and robustness.

**Choice of the $\beta$.** Table 8 studies the effect of redistributing ranks among the $W^Q, W^K$ and $W^V$ matrices, where the adjustment ratio is denoted by $\beta$. We evaluate LLaMA-7B under different compression ratios from 20% to 50% on WikiText-2. The results indicate that an appropriate choice of $\beta$ significantly improves performance compared with the Basis Sharing baseline. In particular, $\beta = 0.3$–$0.4$ consistently yields the lowest PPL across different settings. For example, at 30% compression, D-Rank achieves PPL of 8.87 when group size is 4 compared to PPL of 9.18 for Basis Sharing; at 40% compression, $\beta = 0.35$ gives a PPL of 11.98, clearly better than 12.58 from Basis Sharing. These results show that shifting part of the rank budget from $W^Q, W^K$ to $W^V$ helps the model preserve more informative representations, and that a moderate redistribution of $\beta$ around 0.3–0.4 is most effective.

**Hardware performance of throughput.** Figure 4 reports the throughput of LLaMA-7B under different compression ratios ranging from 20% to 50%. As shown in the figure, all compressed models surpass the dense baseline in terms of tokens processed per second, and the improvement becomes more pronounced as the compression ratio increases. Notably, D-Rank consistently achieves the highest throughput among all approaches. For instance, at 50% compression, our approach reaches nearly 2,200 tokens/sec, which exceeds both SVD-LLM and Basis Sharing and offers more than a 60% gain over the dense model. These results confirm that D-Rank not only preserves accuracy but also brings substantial acceleration benefits in real inference scenarios.

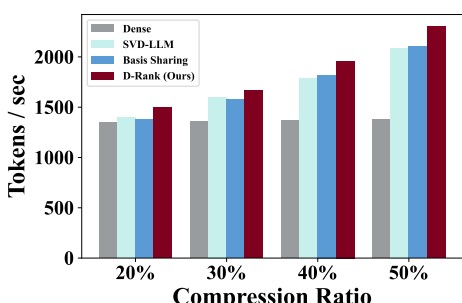

Figure 4: Throughput of dense LLaMA-7B model and the compressed model with Basis Sharing baseline and D-Rank under compression ratios from 20% to 50%.

## 5 CONCLUSION

In this paper, we present **D-Rank**, a novel SVD-based compression framework for large language models. Unlike conventional SVD-based methods, D-Rank dynamically allocates retained ranks for weight matrices across layers to preserve critical information by introducing a novel metric called effective rank to measure weight matrices' information density. By jointly balancing rank distribution across attention layers according to the effective rank of $W^Q, W^K, W^V$, our method achieves a better compression performance. Extensive experiments on different architectures and scales demonstrate that D-Rank consistently reduces perplexity and improves zero-shot reasoning accuracy under 20–50% compression. Moreover, D-Rank remains robust across random seeds and can be seamlessly combined with LoRA fine-tuning to further enhance performance. Overall, D-Rank establishes a practical and effective approach for deploying compression on LLMs.

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

## A CLAIM OF LLM USAGE

In this work, large language models (LLMs) were used solely as a general-purpose writing assistant. Their role was limited to correcting grammar, fixing typographical errors, and polishing the language for clarity and readability.

## B APPENDIX

### B.1 RELATED WORK

**Large Language Model (LLM) Compression.** LLMs typically contain billions of parameters, making traditional training-based compression techniques impractical due to the high computational cost. To alleviate this, post-training compression methods have been widely explored, mainly falling into three major categories: knowledge distillation, pruning, and quantization. Knowledge distillation (KD) (Hinton et al., 2015; Jiao et al., 2020) compresses LLMs by training a smaller student model to mimic the behavior of a larger teacher model. The student learns from the teacher's logits or intermediate representations, thereby reducing the parameter count and inference cost while aiming to preserve performance. However, recent studies (Zhong et al., 2024; Di Palo et al., 2025; Agarwal et al., 2024) have shown that student models often exhibit limited generalization capability compared to their teachers. Pruning removes redundant weights or channels from the original model to produce a sparse subnetwork (Ashkboos et al., 2024a; Sun et al., 2024; Zhang et al., 2024). Unstructured pruning method sets individual weights to zero (Frantar & Alistarh, 2023), while structured pruning removes entire channels or attention heads (An et al., 2024; Wang et al., 2025c; Ling et al., 2024). Although pruning reduces memory and computation, many pruning schemes require retraining, second-order information, or manual sparsity tuning, and they often suffer from performance degradation especially at high sparsity levels (Chen et al., 2021). Quantization reduces model size by representing weights and activations with lower-bit precision such as 8-bit, 4-bit, or even 1–2 bits (Frantar et al., 2022; Zhao et al., 2025). This significantly lowers memory usage and enables faster inference. However, aggressive low-bit quantization (such as 1–2 bits) can introduce substantial accuracy drops (Chee et al., 2023; Li et al., 2025), and quantization-aware training (QAT) requires large datasets and heavy computation (Chen et al., 2025; Liu et al., 2024a), limiting its practicality.

**SVD-based LLM Compression.** Singular Value Decomposition (SVD) reduces matrix dimensionality by truncating the smallest singular values and factorizing the original matrix into three smaller low-rank matrices that approximate it (Golub et al., 1987). SVD-based compression for large language models (LLMs) can simultaneously preserve semantic information and reduce the number of parameters, while allowing the accuracy drop to be controlled. Early studies such as (Denton et al., 2014) demonstrated that applying SVD to convolutional neural networks (CNNs) can substantially accelerate inference without sacrificing accuracy. Building on this idea, (Ben Noach & Goldberg, 2020) applied truncated SVD to BERT-base to obtain an optimal low-rank approximation, which provided high-quality initialization for fine-tuning. However, conventional SVD-based compression assumes all parameters are equally important (Hua et al., 2022), and typically requires fine-tuning after compression to recover performance. To address this limitation, (Hsu et al., 2022) proposed the FWSVD method, which integrates Fisher information into the low-rank decomposition objective to better align the decomposition with task-specific loss. Yet, FWSVD only considers weight importance and overlooks activation outliers or distributional shifts. To mitigate this, (Yuan et al., 2025) introduced ASVD, which preprocesses weights using activation distributions and incorporates outlier influence before performing SVD. Nevertheless, ASVD does not update model parameters after truncation. More recently, (Wang et al., 2025b) presented SVD-LLM, which improves compression efficiency by employing truncation-aware data whitening to align singular values with compression loss and introducing layer-wise closed-form updates. Moreover, Dobi-SVD (Qinsi et al., 2025) introduces a differentiable truncation mechanism combined with theoretical analysis and a weight update formulation, which significantly improves performance under high compression ratios. ResSVD (Bai et al., 2025) leverages the residual matrix generated during the SVD truncation process to reduce truncation errors, and compresses only the latter layers of the model to avoid error accumulation. Despite these advances, most existing studies such as (Yuan et al., 2025; Wang et al., 2025b) focus on compressing and recovering individual layers of large language models, or rely on memory-intensive techniques such as training or backpropagation (Qinsi et al., 2025). However, little work has explored

the compressibility relationships across different layers, and the variation in compressibility among different layer groups remains largely underexplored.

**Parameter Sharing.** Model compression through parameter sharing achieves size reduction by reutilizing weight matrices across multiple layers. (Dehghani et al., 2019) proposed the Universal Transformer, all layers share the same set of parameters, akin to the RNNs, leading to significant parameter reduction. (Reid et al., 2021) categorizes the parameters into attention-related and feedforward-related groups for transformer-based models. These parameters are shared within their respective groups, thereby reducing the overall parameters count while retaining model adaptability. Selective weight sharing is applied to a subset of layers by (Takase & Kiyono, 2023), rather than across all layers. Unlike traditional weight sharing, (Xiao et al., 2019); (Bhojanapalli et al., 2022) explores sharing attention scores across layers. It crucially reduces computational and memory overhead. (Hay & Wolf, 2024) introduce a novel framework, named Dynamic Tying, where reinforcement learning is used to automatically identify optimal layer-wise parameter sharing patterns during training.

### B.2 PERFORMANCE WITH DIFFERENT SEEDS TO SELECT THE CALIBRATION DATA FOR COMPRESSION.

Figure 5 compares the perplexity of different methods on LLaMA-7B when using WikiText-2 as calibration data under varying random seeds. We observe that the performance of both SVD-LLM and Basis Sharing fluctuates with the choice of seed, while D-Rank consistently achieves lower PPL across all settings. For instance, at seed 13, D-Rank obtains 7.45 compared to 7.9 for SVD-LLM and 7.7 for Basis Sharing, and this advantage remains evident even at larger seeds such as 512 and 1024. These results demonstrate that our approach is not only superior in average performance but also more robust to randomness in calibration data selection.

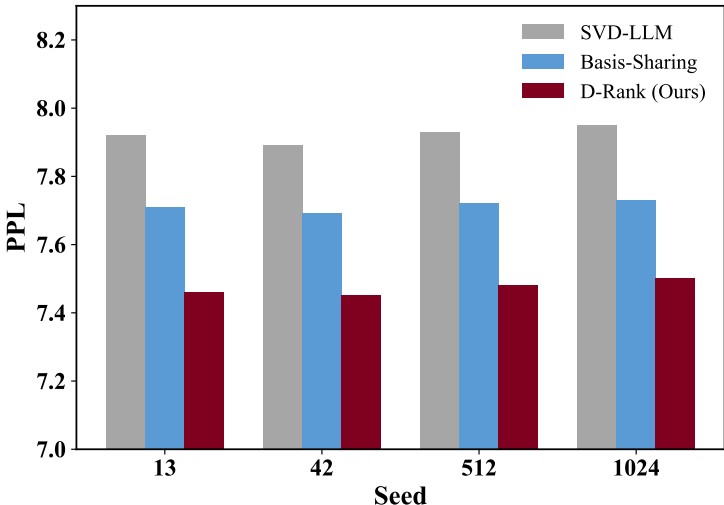

Figure 5: Comparison of PPL with baselines on LLaMA-7B model when selecting the calibration data from Wikitext-2 with different seeds to compute $\mathcal{S}$

### B.3 RANK ALLOCATION VIA LAGRANGE MULTIPLIERS

Let $k_g$ be the retained rank for group $g \in \{1, \ldots, G\}$, $\mathcal{R}_{\text{eff}}(g)$ the effective rank (information measure), $\omega$ the parameter cost per unit rank for group $g$, and $\mathcal{T}_{\text{budget}}$ the total rank cost budget as defined in Section 3.2. We minimize the loss under a budget constraint:

$$\min_{k_1,\ldots,k_G} \quad \ell_{\text{total}} = \sum_{g=1}^{G} \frac{\mathcal{R}_{\text{eff}}(g)}{k_g} \quad \text{s.t.} \quad \sum_{g=1}^{G} k_g\,\omega = \mathcal{T}_{\text{budget}} \tag{13}$$

The Lagrangian is

$$\mathcal{F}(\{k_g\}, \lambda) \;=\; \sum_{g=1}^{G} \frac{\mathcal{R}_{\text{eff}}(g)}{k_g} \;+\; \lambda \left( \sum_{g=1}^{G} k_g \, \omega - \mathcal{T}_{\text{budget}} \right) \tag{14}$$

Setting the derivative w.r.t. each $k_g$ to zero:

$$\frac{\partial \mathcal{F}}{\partial k_g} \;=\; -\frac{\mathcal{R}_{\text{eff}}(g)}{k_g^2} \;+\; \lambda \, \omega \;=\; 0 \;\;\Longrightarrow\;\; k_g \;=\; \sqrt{\frac{\mathcal{R}_{\text{eff}}(g)}{\lambda \, \omega}} \tag{15}$$

Hence the optimal ranks follow the proportionality

$$k_g \;\propto\; \frac{\sqrt{\mathcal{R}_{\text{eff}}(g)}}{\sqrt{\omega}} \tag{16}$$

Let $C$ be the proportionality constant. Using the budget constraint,

$$\sum_{g=1}^{G} k_g \, \omega \;=\; \sum_{g=1}^{G} \left( C \, \frac{\sqrt{\mathcal{R}_{\text{eff}}(g)}}{\sqrt{\omega}} \right) \omega \;=\; C \sum_{g=1}^{G} \sqrt{\mathcal{R}_{\text{eff}}(g) \, \omega} \;=\; \mathcal{T}_{\text{budget}} \tag{17}$$

so we have:

$$C \;=\; \frac{\mathcal{T}_{\text{budget}}}{\sum_{j=1}^{G} \sqrt{\mathcal{R}_{\text{eff}}(j) \, \omega}} \tag{18}$$

Substituting $C$ back yields the final closed-form allocation:

$$k_g \;=\; \frac{\mathcal{T}_{\text{budget}}}{\sum_{j=1}^{G} \sqrt{\mathcal{R}_{\text{eff}}(j) \, \omega}} \cdot \frac{\sqrt{\mathcal{R}_{\text{eff}}(g)}}{\sqrt{\omega}} \tag{19}$$

*Interpretation.* Equation 16-19 show that groups with larger information content $\mathcal{R}_{\text{eff}}(g)$ receive higher ranks, whereas groups with higher parameter cost $\omega$ receive fewer ranks, all under the fixed budget $\mathcal{T}_{\text{budget}}$.