# OpenReview forum: "Layer-wise dynamic rank for compressing large language models"
_ICLR.cc/2026/Conference — Submitted to ICLR 2026_

### Official Review · Reviewer_NDjg · 2025-10-29

**Soundness:** 2
**Presentation:** 3
**Contribution:** 2
**Rating:** 2
**Confidence:** 4

**Summary:**

This paper proposes D-Rank, a SVD-based compression work for LLM that dynamically allocates compression ranks across layers based on their information density, rather than applying uniform compression ratios.  Given the limited tech contribution, and insufficient experiments, this paper should be improved before it can be published.

**Strengths:**

well written paper, easy to follow.

**Weaknesses:**

1. Missing citation and limited novelty. The paper's core contribution—dynamic rank allocation for LLM compression—is not new. Hua et al. (2023) "Dynamic Low-rank Estimation for Transformer-based Language Models" (EMNLP 2023 Findings) directly addresses dynamic rank estimation for transformers. The authors cite the 2022 work from the same group but omit their 2023 follow-up on the exact same problem.  This makes the novelty claims appear overstated.
2. The core technical contribution—using spectral entropy-based effective rank to measure information density—lacks proper justification.  The paper provides no explanation for why spectral entropy is the optimal metric, or does it compare against alternatives like stable rank, nuclear norm, Fisher information (FWSVD), or activation-based importance (ASVD). Table 1 shows effective rank values vary across layers, but doesn't demonstrate these values actually correlate with optimal compression performance. The mathematical formulation (Equations 1-2: normalizing squared singular values into a probability distribution, then exponentiating the Shannon entropy) appears arbitrary without theoretical grounding.
3. Unclear and Potentially Unfair Experimental Settings. The experimental setup is poorly specified, creating uncertainty about fair comparison. For example, Table 3 shows FWSVD with perplexity 18156 at 40% - is this without fine-tuning while D-Rank gets updates? Looking at Table 3 (LLaMA-7B results), the numbers for FWSVD and basic SVD are catastrophically bad (perplexities in the thousands), while ASVD, SVD-LLM, Basis Sharing, and D-Rank are reasonable. This suggests: A) FWSVD and SVD are evaluated zero-shot (no recovery procedure); B) ASVD, SVD-LLM, Basis Sharing, D-Rank get some form of correction/update

**Questions:**

1. Why is spectral entropy the right metric for information density?
2. Computing spectral entropy requires all singular values. How numerically stable is this for large matrices? Any regularization?

---

> ### Author Response · Authors · 2025-11-26
>
> **[Comparing with RankDyna]**
>
> We thank the reviewer for the comment and for comparing our work with RankDyna.
> However, we identify the difference of the two works from several aspects as follows:
> The goal of RankDyna is reducing memory and computation during fine-tuning to improve learning efficiency and generalization. It regards the rank as an adaptable variable during fine-tuning, controlling how many ranks each layer adapts during the fine-tuning epoch. However, D-Rank, in contrast, derives an *information density guided effective rank* for layer-wise dynamic ranks under a global compression budget, targeting deployment efficiency rather than training dynamics.
> From a methodological perspective, RankDyna operates during the fine-tuning stage by, for each singular component of a weight matrix, estimating the change in loss via a Taylor expansion when that singular component is removed. This procedure is useful for assessing the contribution of each singular direction during training. However, it does not capture the overall information content of the matrix, and therefore cannot indicate how many singular directions are sufficient to preserve the matrix’s essential structure. D-Rank solves a **single constrained optimization problem** with an explicit analytic solution with effective rank which can indicate how many ranks do we need to recover a specific matrix , providing theoretical interpretability, reproducibility, and zero training overhead.
> In terms of implementation stage, RankDyna  operates **online during fine-tuning**, requiring gradient or activation-based loss and iterative rank updates. However, D-Rank is a **post-training analytic allocation** that needs no retraining, gradients, or activation sampling, only the pre-trained model’s singular values.
>
> In general, RankDyna optimizes training efficiency, while D-Rank optimizes post-training compression and deployment. They operate at different stages, pursue different goals, and rely on fundamentally different theoretical formulations and allocation methods.

---

> ### Author Response · Authors · 2025-11-26
>
> **[Comparison of effective rank with other metrics]**
>
> We appreciate the reviewer’s request for theoretical justification of the spectral-entropy metric.
> The use of spectral entropy is not arbitrary; it follows a clear information-theoretic rationale with solid mathematical grounding [1].  The effective rank measures how many dimensions of a matrix are truly useful for dimensionality reduction or principal component analysis. Although this concept originates from statistics and signal processing, its generality and well-defined mathematical properties support its use for model compression.
>
> When the squared singular values are normalized as
> $p_i = \sigma_i^2 / \sum_j \sigma_j^2$,
> their Shannon entropy is
>
> $$
> H(p) = -\sum_i p_i \log p_i .
> $$
>
> Its exponential,
>
> $$
> R_{\mathrm{eff}} = \exp(H(p)),
> $$
>
> corresponds to the participation ratio widely used to quantify the effective number of active degrees of freedom.
> Therefore, according to [1], a layer with uniform singular values (flat spectrum) yields high entropy → high information diversity → requires higher rank, whereas a layer with a sharply decayed spectrum yields low entropy → low intrinsic dimensionality → can be compressed more aggressively.
>
>
> Our proposed spectral-entropy effective rank measures information diversity rather than absolute spectral energy or task-specific sensitivity.
>
> In contrast, the stable rank defined as
> $\|W\|_F^2 / \|W\|_2^2$
> depends mostly on the largest singular value and cannot distinguish spectra with similar total energy.
> Stable rank only reflects how many directions have energy comparable to the dominant singular direction.
>
> To illustrate this, we computed the stable rank of the Q and V matrices in LLaMA-7B (group\_size = 2):
>
> Stable rank of V:
>   [4.9, 15.6, 18.2, 15.0, 16.0, 21.7, 23.9, 20.4, 18.1, 16.9, 14.3, 12.8, 13.7, 11.8, 11.8, 9.4]
>
> Stable rank of Q:
>   [2.0, 1.3, 1.5, 1.6, 1.8, 1.8, 1.9, 1.8, 1.7, 1.7, 1.6, 1.7, 1.6, 1.7, 1.6, 1.7]
>
> We observe that the stable ranks of Q and V are in a similar magnitude range, which fails to capture the substantial information disparity between V and Q/K. Thus, stable rank does not provide insight that would motivate balancing-based rank reallocation.
>
> The nuclear norm, $\|W\|_* = \sum_i \sigma_i$, measures total magnitude but lacks scale invariance, making cross-layer comparison unreliable.
>
> ASVD determines the number of retained singular values experimentally, relying on empirical validation rather than a theoretical metric or analytic rule. FWSVD requires estimating second-order gradient statistics (Fisher Information Matrix),  which incurs extremely high computational cost.
>
> In contrast, D-Rank relies solely on the singular-value spectrum and does not require activation statistics, gradients, or second-order information. Table 3 shows that D-Rank consistently outperforms ASVD and FWSVD under SVD-based compression:
>
> | Compression Ratio | FWSVD  | ASVD  | D-Rank |
> |-------------------|--------|-------|----------------|
> | 20%               | 1727   | 11.14 | **7.45**       |
> | 30%               | 20127  | 51    | **8.97**       |
> | 40%               | 18156  | 1407  | **11.99**      |
> | 50%               | 24391  | 13588 | **19.82**      |
>
> Overall, the spectral-entropy effective rank yields a task-agnostic, smooth, and theoretically grounded measure of the effective degrees of freedom in each layer, enabling analytic and globally consistent rank allocation without any retraining.
>
>
> [1] O. Roy and M. Vetterli. *The effective rank: A measure of effective dimensionality.* In EUSIPCO 2007, pages 606–610. IEEE, 2007.

---

> ### Author Response · Authors · 2025-11-26
>
> **[Questions with experiment settings]**
>
> We thank you for pointing out your concern. We clarify that all methods were evaluated under the same compression and evaluation protocol, and no fine-tuning or gradient updates were applied to D-Rank. Our D-Rank results are obtained purely through post-training analytic rank allocation, without any retraining or recovery procedure.
>
> Regarding the seemingly extreme perplexity of FWSVD (≈18,156 at 40% compression in Table 3): This number is **exactly the same as that reported in the original SVD-LLM paper** [1], from which we reproduced the baseline following their open-sourced  implementation and hyperparameters. Hence, the large perplexity gap does **not** come from unfair settings or additional updates in D-Rank. We will explicitly clarify this in the revised version to avoid misunderstanding.
>
> [1] Wang, Xin, et al. "Svd-llm: Truncation-aware singular value decomposition for large language model compression." arXiv preprint arXiv:2403.07378 (2024).

---

> ### Author Response · Authors · 2025-11-26
>
> [Stability of SVD]
>
> Mathematically, the numerical stability of an SVD decomposition is largely determined by the condition number of the matrix. When the condition number is excessively large, the SVD becomes numerically unstable. To verify this, we computed and printed the condition numbers of the Q, K, and V matrices in LLaMA-7B (with group size = 4). We observe that none of the matrices exhibit condition numbers beyond the range that floating-point arithmetic can reliably handle. Therefore, throughout our entire SVD-based compression pipeline, no numerical instability issues arise. We also note that no regularization was applied during the SVD decomposition process.
>
> | Group ID |   Q condition number   |   K condition number   |   V condition number   |
> |----------|-----------------|----------------|----------------|
> | 0        | 4438.57         | 5593.59        | 398.11         |
> | 1        | 1446.33         | 1344.25        | 212.09         |
> | 2        | 766.44          | 783.58         | 136.08         |
> | 3        | 573.46          | 621.99         | 131.80         |
> | 4        | 560.17          | 596.54         | 120.52         |
> | 5        | 514.02          | 563.16         | 209.00         |
> | 6        | 403.84          | 479.87         | 312.76         |
> | 7        | 442.80          | 522.06         | 145.37         |

---

### Official Review · Reviewer_XMLc · 2025-10-31

**Soundness:** 3
**Presentation:** 3
**Contribution:** 2
**Rating:** 6
**Confidence:** 3

**Summary:**

This paper enhances SVD-based methods by assigning different compression ratios to different layers, which improves model accuracy. Leveraging an "effective rank" metric, the authors assign a higher rank to layers with higher information density. The superiority of this approach is validated on numerous models.

**Strengths:**

1. The motivation is clear, and the paper is well-written.

2 .The results of hardware performance are provided.

3. A combination with LoRa fine-tuning is provided.

**Weaknesses:**

1. A comparison with SVD-LLM v2 is missing; that method also assigns different compression ratios to different layers.

2. Experiments on 70B models should be added.

3. Figure 1 should be improved, and a description of the image should be added to the caption.

4. Separate ablation studies for the "dynamic rank" and "balancing" components are lacking.

**Questions:**

Please see the weaknesses above, and:

1. Please explain the difference between the "effective rank" metric established in this paper and the loss-based metric used in SVD-LLM v2. which is better?

---

> ### Author Response · Authors · 2025-11-25
>
> **[Fair comparison with SVD-LLM V2]**
>
> We appreciate the reviewer’s suggestion. Since the SVD-LLM V2 paper has not released its implementation, and because the paper introduces two major innovations: (1) layer-wise rank compression and (2) an additional calibration-set–based refinement step, so we are unable to isolate the contribution of each component. The paper does not provide ablation studies that evaluate these two techniques separately. Therefore, we cannot fairly compare D-Rank with “SVD-LLM V2 using only layer-wise rank compression.” Moreover, the calibration-set refinement technique proposed in SVD-LLM V2 is orthogonal and can in principle be incorporated into D-Rank to further improve performance.
>
> **[Experiments for larger models]**
>
> We appreciate the reviewer’s suggestion for extending D-Rank to larger models such as 70B. We are currently extending our experiments to larger-scale models to further validate scalability.  However, existing baselines such as SVD-LLM and Basis Sharing have not reported 70B results, so reproducing comparable baselines requires additional computation and extra effort. We are actively working on this extension and will include larger model  results in the revised version once available. We sincerely thank the reviewer for the understanding.
> But now we have done the experiments of LLaMA-65B model of 40% compression ratio with the wikitext-2 dataset as calibration data and evaluation data. D-Rank can achieve 6.26 PPL and while Basis sharing can achieve only 6.42 PPL. We will complement more experiments of larger model in our final version.
>
> **[Improvement for Figure 1]**
>
> We thank the reviewer for this helpful suggestion. We are currently improving the visual quality of Figure 1 and will add a more detailed and informative caption describing the figure contents in the revised version.
>
> **[Experiments for dynamic rank and balancing]**
>
> We thank the reviewer for pointing out the need to disentangle the contributions of the two components. We have added ablation experiments on LLaMA-7B, using C4 as both the calibration corpus and evaluation dataset,  with 4 layers grouped together following the Basis Sharing setting. The results are as follows under 20% compression ratio:
>
> | Method                                   | PPL (C4) |
> |:----------------------------------------|:--------:|
> | Basis Sharing                           | 11.42    |
> | **D-Rank** | **10.78** |
> | **dynamic rank**    | 11.15    |
> | **balancing / attention reallocation**  | 11.03    |

---

> ### Author Response · Authors · 2025-11-26
>
> We thank the reviewer for the comment.
>
> SVD-LLM v2 determines the layer-wise rank based on truncation loss, which is more heuristic and experimental. The rank to be preserved is chosen as follows: for each layer $i$, it computes the reconstruction error on a calibration set as
>
> $$
> L_i = \| W_i X_i - W_i^{(r)} X_i \|_F^2 ,
> $$
>
> where $W_i^{(r)}$ is the weight matrix truncated to $r$ singular values. Layers that incur larger losses under the same compression ratio are assigned higher ranks, while layers with smaller losses receive lower ranks. In other words, the rank allocation is loss-driven, which heuristically balances per-layer reconstruction errors.
>
> In contrast, D-Rank does not rely on reconstruction loss or activation data. Instead of focusing on only Frobenius-norm loss, D-Rank formulates rank allocation as an optimization problem guided by information density.  We compute each layer’s spectral-entropy effective rank to quantify its information diversity, and derive a closed-form global allocation under a total parameter-budget constraint.  Thus, SVD-LLM v2 performs empirical loss equalization,  whereas D-Rank performs analytic information maximization, a conceptually and computationally distinct approach.
>
> Moreover, the two methods differ not only in objectives but also in how ranks are allocated.
>
> SVD-LLM v2 performs loss-based empirical balancing: after estimating each layer’s reconstruction loss
>
> $$
> L_i = \| W_i X_i - W_i^{(r)} X_i \|_F^2 ,
> $$
>
> on calibration data, it heuristically adjusts ranks so that high-loss layers receive larger $r_i$ and low-loss layers receive smaller $r_i$. This procedure is local and provides no closed-form guarantee of global optimality.
>
> In contrast, D-Rank introduces a principled, analytic allocation rule. We compute each layer’s spectral-entropy effective rank
>
> $$
> R_{\mathrm{eff}} = \exp(H(p)),
> $$
>
> which reflects its intrinsic information content and diversity, and derive layer ranks $k_i$ from a Lagrange-constrained optimization that enforces a total parameter budget. This results in a closed-form, globally consistent allocation, without requiring activations, calibration data, or retraining.

---

### Official Review · Reviewer_51KQ · 2025-11-01

**Soundness:** 2
**Presentation:** 2
**Contribution:** 1
**Rating:** 2
**Confidence:** 4

**Summary:**

The paper proposes D-Rank, a layer-wise dynamic rank allocation framework for SVD-based compression of large language models. It introduces an “effective rank” metric based on spectral entropy to measure the information density of each layer’s weight matrices and uses a Lagrange-multiplier optimization to allocate ranks proportionally. The method also redistributes ranks among matrices and adapts to grouped-query attention (GQA) models such as LLaMA-3.

**Strengths:**

1. The paper identifies layer-wise heterogeneity in information density and attempts to address it with an adaptive rank allocation strategy.
2. The effective rank metric provides a simple quantitative measure for weight information, improving interpretability of rank selection.

**Weaknesses:**

- The approach mainly extends existing SVD-LLM and Basis-Sharing ideas by replacing uniform rank selection with a simple entropy-based allocation. The Lagrange-multiplier optimization and “effective rank” metric are straightforward and do not represent a substantial algorithmic advance.

- The method depends on activation statistics and per-layer SVD computation, adding complexity without clear end-to-end throughput gains in real inference systems. It lacks compatibility with prefix-cache frameworks (e.g., SGLang) or kernel-fused inference, which severely limits deployment feasibility.

- Although throughput is reported, experiments are small-scale and ignore key system factors such as multi-GPU memory partitioning and cache reuse. The cost of computing effective ranks may offset compression gains.

- The entropy-based rank metric lacks clear connection to model generalization or reconstruction fidelity. No analysis links effective rank distributions to actual representational importance.

**Questions:**

- How does the additional SVD and activation-statistic computation affect total compression and inference time?

- Does “effective rank” correlate with layer importance beyond empirical intuition? A deeper justification or visualization would help.

---

> ### Author Response · Authors · 2025-11-25
>
> **[Straightforward effective rank metric with no substantial algorithmic advance]**
>
> We thank the reviewer of the comment. We agree that our framework builds upon the general SVD-based compression pipeline, but the motivation of this work is not a mere variant of SVD-LLM,  it originates from a key empirical observation: Different  layers’ matrices exhibit highly non-uniform information densities, and using a uniform or heuristic rank allocation as in prior work such as SVD-LLM and Basis sharing may not be good enough and leads to suboptimal compression and severe performance imbalance across layers.
> To address this structural issue, we introduce an information density evaluation metric that quantifies each layer’s intrinsic information content via spectral-entropy effective rank, and derive a closed-form global allocation rule through Lagrange optimization under a total parameter budget constraint according to the effective rank metric which expresses the information content of the matrix of each layer.
> This transforms rank assignment from an empirical heuristic into a principled analytic process, achieving both interpretability and parameter efficiency without retraining.  Although some existing works such as WeLore and ASVD also adopt layer‐wise varying compression ratios, our method is more theoretically grounded and interpretable compared with approaches like ASVD, which requires running data-dependent experiments to determine dynamic ranks. In contrast, WeLore normalizes and sorts all singular values across layers and prunes them globally, but this strategy does not account for the distribution of information content within each matrix. Consequently, under high compression ratios, WeLore performs worse than D-Rank. We provide the PPL of D-Rank compared to WeLore and ASVD baseline on Wikitext-2.
>
> | Compression Ratio | ASVD  | D-Rank |
> |-------------------|-------|----------------|
> | 20%               | 11.14 | **7.45**       |
> | 30%               | 51  | **8.97**       |
> | 40%               | 1407 | **11.99**      |
> | 50%               | 15358 | **19.82**      |
>
> | Compression Ratio | WeLore | D-Rank |
> |-------------------|--------|--------|
> | 10%               | 6.03   | 6.36   |
> | 15%               | 9.72   | 7.02   |
> | 33%               | 932.06 | 10.01  |
>
>
>  **[limitation of deployment feasibility]**
>
> We thank the reviewer for this comment. We clarify that the activation statistics mentioned in the paper are used only once (in line with SVD-LLM and Basis Sharing baseline) during the post-training compression stage to estimate layer information density and compute the spectral-entropy effective rank. After rank allocation and re-factorization, the compressed model is saved in a static low-rank format. During inference, no activation statistics collection, SVD computation, or additional processing is required, so the model runs identically to standard SVD-LLM or Basis Sharing models. Therefore, D-Rank introduces no runtime overhead, and remains fully compatible with existing prefix-cache and kernel-fused inference frameworks (e.g., SGLang, vLLM, TensorRT-LLM).
>
> **[Experiments of model scales]**
>
> We thank you for this comment and clarify a key point: the computation of effective ranks is performed only once offline during the post-training compression stage, not during inference. This process involves a single SVD per layer to obtain singular values, after which ranks are analytically allocated and the compressed weights are stored as static low-rank matrices. During inference, the model behaves exactly like any other SVD-compressed model (e.g., SVD-LLM) and requires no additional computation, memory, or synchronization across GPUs. Therefore, the effective-rank estimation cost has no impact on inference runtime throughput or system-level deployment such as multi-GPU or cache.
> Moreover, we have conducted different scales of models in Table 6.
> | Method        |   7B   |   13B  |   30B  |
> |---------------|--------|--------|--------|
> | SVD           | 20061  | 946.31 | 54.11  |
> | FWSVD         | 1630   | OOM    | OOM    |
> | ASVD          | 11.14  | 6.74   | 22.71  |
> | SVD-LLM       | 7.94   | 6.61   | 5.63   |
> | Basis Sharing | 7.75   | 6.47   | 5.47   |
> | **D-Rank (Ours)** | **7.45** | **6.30** | **5.33** |

---

> ### Author Response · Authors · 2025-11-25
>
> **[Relation of effective rank with model layer importance]**
>
> We thank the reviewer for this comment.
>
> **Theoretical:**
> The use of spectral entropy is not arbitrary but follows a clear information-theoretic rationale and has solid mathematical grounding [1]. In our work, the effective rank measures how many dimensions of a matrix are truly useful for dimensionality reduction or principal component analysis. Although the original definition comes from statistics and signal processing, its generality and well-defined mathematical properties justify its use in model compression.
> When the squared singular values are normalized as
> $p_i = \sigma_i^2 / \sum_j \sigma_j^2$,
> their Shannon entropy is
> $$
> H(p) = -\sum_i p_i \log p_i .
> $$
> Its exponential,
> $$
> R_{\mathrm{eff}} = \exp(H(p))
> $$
> corresponds to the participation ratio widely used in signal processing to quantify the effective number of active degrees of freedom. Therefore, according to [1], a layer with uniform singular values (flat spectrum) has high entropy → high information diversity → needs higher rank, while a layer with sharply decayed spectrum has low entropy → low intrinsic dimensionality → can be compressed more aggressively.
>
> **Experimental:**
> To experimentally evaluate this, we conducted an ablation on the LLaMA-7B model, where two layers were grouped following the Basis Sharing setup. For example, under the static rank allocation used in Basis Sharing, each Q/K/V group retains a fixed 2,457 singular values. In contrast, D-Rank dynamically allocates ranks per group according to the spectral-entropy–based effective rank.
> Using the effective rank to determine per-group rank consistently yields **smaller Frobenius-norm reconstruction error** than Basis Sharing. We computed the Frobenius-norm loss of the Q, K, and V matrices on LLaMA-7B with `group_size = 2` and **20% compression**, and observed a clear reduction in reconstruction error when using D-Rank:
>
> | Matrix | Basis-Sharing F-norm Loss | D-Rank F-norm Loss | Relative Reduction |
> |:------|---------------------------:|-------------------:|-------------------:|
> | **V** | 3,128,481,266              | **2,228,811,324**  | ↓ 28.8%            |
> | **K** | 3,101,934,300              | **2,724,827,996**  | ↓ 12.1%            |
> | **Q** | 2,924,171,431              | **2,466,980,789**  | ↓ 15.6%            |
>
> Overall, the total reconstruction loss summed over all layers decreases substantially with D-Rank.
> This demonstrates that entropy-based dynamic rank allocation better preserves layer-wise information, confirming that the effective rank meaningfully reflects representational importance rather than being a purely abstract metric. Furthermore, corresponding generalization results across different model scales are shown in Tables 5–6.
>
> [1] O. Roy and M. Vetterli. *The effective rank: A measure of effective dimensionality.* In EUSIPCO 2007, pages 606–610. IEEE, 2007.
>
>
> **[Effect of additional SVD and activation-statistic computation on total compression and inference time]**
>
> Thanks to the reviewer to this comment.
> We conducted an experiment to measure the additional SVD and activation-statistic computation overhead. On the LLaMA-7B model, this process took approximately 18.9 minutes in total. However, for D-Rank as well as the SVD-LLM and Basis Sharing baselines, this computation is a one-time offline operation performed only during compression; it is not required at inference time.
> Since D-Rank follows exactly the same SVD and activation-statistic procedure as the baselines, their preprocessing times are identical. For reference, the inference time of a 40 %-compression LLaMA-7B model with Basis sharing  is about 3 minutes 13 seconds on Wikitext-2 dataset, and the inference time of a 40 %-compression LLaMA-7B model with D-Rank  is about 2 minutes 58 seconds on Wikitext-2 dataset. Moreover, the inference time of a 30%-compression LLaMA-7B model with Basis sharing  is about 3 minutes 51 seconds on Wikitext-2 dataset, and the inference time of a 30 %-compression LLaMA-7B model with D-Rank  is about 3 minutes 40 seconds on Wikitext-2 dataset.

---

### Official Review · Reviewer_F3JZ · 2025-11-02

**Soundness:** 3
**Presentation:** 3
**Contribution:** 2
**Rating:** 6
**Confidence:** 3

**Summary:**

The paper targets the rank-allocation problem in SVD-based LLM compression and proposes D-Rank. The method first computes an effective rank for each group of layers, then uses a Lagrange-multiplier formulation to assign larger ranks to groups with higher effective rank under a fixed global budget. On top of this, it introduces an attention-specific rebalancing step that shifts rank from query/key to value matrices via a hyperparameter, motivated by the empirical observation that value matrices carry higher effective rank. The method also covers the grouped-query attention case in LLaMA-3, where layer grouping degrades compression, by falling back to group size 1 while still applying dynamic rank allocation. Experiments across multiple LLMs and datasets show consistent improvements over existing SVD-based baselines, and ablations confirm the contribution of each component as well as compatibility with LoRA fine-tuning.

**Strengths:**

* The rank allocation problem is an important and interesting direction to explore.
* The paper is clearly written, with the methodology and experimental setup explained in a smooth and logical way.
* Using an entropy-based effective-rank metric is a reasonable choice for quantifying information density across layers/groups.
* The experiments are comprehensive, and the ablation studies help isolate the contribution of each component.

**Weaknesses:**

* The novelty is somewhat limited. Structure-aware and adaptive low-rank allocation compression have already been explored in recent work (for example, see [1]).
* The attention-specific reallocation is empirically effective, but the large gap between value matrices and query/key matrices suggests that grouping them together may not be ideal. It would be helpful to discuss whether an alternative grouping strategy could address this mismatch and potentially remove the extra hyperparameter $\beta$?

[1] Hua et al; Dynamic Low-rank Estimation for Transformer-based Language Models.

**Questions:**

* How sensitive is the estimated effective rank to the choice of calibration data? For example, does it change significantly if a different subset or a different source corpus is used?
* How important is the entropy-based effective-rank metric itself? It would be useful to report results with alternative approximation method s (For example, stable rank (Fro. norm over 2-norm) or a simple energy-based threshold such as “95% SV energy”) ?

---

> ### Author Response · Authors · 2025-11-25
>
> **[Comparing with RankDyna]**
>
> We thank the reviewer for the comment and for comparing our work with RankDyna.
> However, we identify the difference of the two works from several aspects as follows:
> The goal of RankDyna is reducing memory and computation during fine-tuning to improve learning efficiency and generalization. It regards the rank as an adaptable variable during fine-tuning, controlling how many ranks each layer adapts during the fine-tuning epoch. However, D-Rank, in contrast, derives an *information density guided effective rank* for layer-wise dynamic ranks under a global compression budget, targeting deployment efficiency rather than training dynamics.
> From a methodological perspective, RankDyna operates during the fine-tuning stage by, for each singular component of a weight matrix, estimating the change in loss via a Taylor expansion when that singular component is removed. This procedure is useful for assessing the contribution of each singular direction during training. However, it does not capture the overall information content of the matrix, and therefore cannot indicate how many singular directions are sufficient to preserve the matrix’s essential structure. D-Rank solves a **single constrained optimization problem** with an explicit analytic solution with effective rank which can indicate how many ranks do we need to recover a specific matrix , providing theoretical interpretability, reproducibility, and zero training overhead.
> In terms of implementation stage, RankDyna  operates **online during fine-tuning**, requiring gradient or activation-based loss and iterative rank updates. However, D-Rank is a **post-training analytic allocation** that needs no retraining, gradients, or activation sampling, only the pre-trained model’s singular values.
>
> In general, **RankDyna optimizes training efficiency**, while **D-Rank optimizes post-training compression and deployment**.
> They operate at different stages, pursue different goals, and rely on fundamentally different theoretical formulations and allocation methods.
>
> **[Extra hyperparameter of β]**
>
> We thank the reviewer for the comment. In our framework, Q, K, and V matrices are not grouped together within the same layer.  Instead, we perform grouping across layers for each matrix type (e.g., two consecutive layers’ Q matrices form one group, and similarly for K and V).
> For each group, we estimate its *effective rank* based on spectral entropy. We observe that Q/K groups generally exhibit lower effective ranks than V groups, indicating that value matrices retain more information diversity.   To maintain the overall parameter budget, we reallocate a small portion of unused rank capacity from Q, K groups to V groups, controlled by a single scalar coefficient β.
> This β determines how much rank “budget” is transferred from Q, K to V and remains fixed across all layers and models. This design ensures balanced information preservation between low effective rank Q, K and high effective rank V components.
> **Regarding why the optimal value of β in our experiments is around 0.3–0.4:** because the effective rank V of the value matrix is relatively high, choosing a β that is too small fails to preserve sufficient information in V. On the other hand, setting β too large drastically reduces the rank of the Q, K matrix, which breaks the correspondence between Q, K and V. As a result, the perplexity of the compressed model can no longer be improved.
>
> **[Sensitivity of effective rank to calibration data]**
>
> We appreciate the reviewer’s question on data dependence.
> The estimated effective ranks indeed vary with different calibration data,
> which is expected since activation statistics and spectral energy distributions reflect data-specific token dynamics.
> However, the overall **layer-wise trend remains consistent**, which indicates that the metric captures intrinsic structural properties rather than dataset noise for specific LLM model.
> For example, on LLaMA-7B model with group size of 4,  when we use **C4** as the calibration data,
> the effective ranks of the value matrices across seven grouped layers are:
> [697.2, 838.3, 1082.1, 1487.5, 1577.0, 1778.8, 1100.7].
> When we use use **WikiText-2**, the corresponding ranks are:
> [580.6, 674.5, 877.1, 1157.2, 1172.8, 1334.8, 846.6].

---

> ### Author Response · Authors · 2025-11-25
>
> [Importance of effective rank metric]
>
> We thank the reviewer for this valuable question. We emphasize that the entropy-based effective rank is not merely another energy-threshold heuristic, but a more general information metric that captures the diversity of singular-value energy distribution in matrices, not just the total retained energy. Stable rank $\frac{\|W\|_F^2}{\|W\|_2^2}$
>  only reflects the ratio between total and maximum energy, making it insensitive to the overall spectral shape. It cannot distinguish between gradual and sharply decayed spectra, and thus fails to capture the diversity of active directions in the representation space.
> Energy-based thresholding (e.g., 95% cumulative energy) requires a manually chosen cutoff, introducing an arbitrary hyperparameter that varies across models and layers. It considers only energy magnitude, not its distribution, and produces unstable rank estimates under different scaling or normalization.
> In contrast, our entropy-based effective rank incorporates the entire singular value spectrum and quantifies the information diversity via Shannon entropy without any tunable thresholds. This makes it theoretically grounded, parameter-free, and more robust for analytic rank allocation.
> To experimentally assess the importance of the entropy-based effective rank, we conducted additional experiments using the LLaMA-7B model, with WikiText-2 serving as both the calibration and evaluation dataset. Two consecutive layers were grouped following the same setting as in Basis Sharing. We replaced our effective rank  metric with two alternatives: Stable rank: $\frac{\|W\|_F^2}{\|W\|_2^2}$ and Energy-based threshold (rank chosen to retain 95% cumulative singular-value energy)
>
> Under the same 20% global compression ratio without balancing, using stable rank for allocation yields a perplexity of 7.69.
> And  95% energy threshold yields 7.80, while our  effective rank achieves the best performance at 7.62. This demonstrates that the effective rank provides a better estimate of layer-wise information diversity than existing alternatives.
>
> In addition, we would like to emphasize that it is precisely the uneven information density across Q, K, and V matrices revealed by the effective rank that motivated our balancing strategy. Thus, although stable rank may have some benefits for SVD-based compression compared to baselines, the uneven distribution of effective rank between Q, K, V matrices motivates us to further balance the ranks of them.
> As shown by the stable-rank and 95% energy-threshold results we computed, the differences between Q/K/V under these alternative metrics are much smaller and in some cases nearly negligible.
> Therefore, using these metrics would not have highlighted the imbalance across Q/K/V, and thus would not have led us to develop the balancing method that further improves performance.
>
> Stable rank of V: [4.9, 15.6, 18.2, 15.0, 16.0, 21.7, 23.9, 20.4, 18.1, 16.9, 14.3, 12.8, 13.7, 11.8, 11.8, 9.4]
>
> Stable rank of Q: [2.0, 1.3, 1.5, 1.6, 1.8, 1.8, 1.9, 1.8, 1.7, 1.7, 1.6, 1.7, 1.6, 1.7, 1.6, 1.7]
>
> 95% energy-based thresholding of V: [815, 1730, 1874, 1842, 1717, 1815, 1977, 2120, 2229, 2280, 2308, 2360, 2491, 2544, 2556, 2417]
>
> 95% energy-based thresholding of Q: [35, 95, 348, 470, 522, 623, 746, 776, 871, 844, 927, 1037, 1276, 1431, 1243, 1015]

---

### Author Response · Authors · 2025-12-02

Our paper proposes D-Rank, a dynamic rank allocation framework tailored for SVD-based compression of large language models. D-Rank introduces an effective-rank metric to quantify the information density of each weight matrix and employs a Lagrange-multiplier optimization to allocate ranks proportionally under a fixed global budget. The method further incorporates attention-specific balancing to reflect the higher representational importance of value matrices and adapts naturally to grouped-query attention architectures used in modern LLMs such as LLaMA-3. Experiments across multiple LLM families and a wide range of compression ratios consistently show improved perplexity, downstream accuracy, and inference throughput over uniform SVD-based baselines.

Across reviewers, there is strong agreement on several key strengths:
The paper tackles an important challenge in SVD-based LLM compression, addressing layer-wise and matrix-wise heterogeneity in information density rather than relying on uniform-rank heuristics.
The effective-rank metric is recognized as intuitive, interpretable, and practically useful, providing a simple and principled way to quantify weight information content for allocating compression budgets.
The methodology is clear and well-structured, with reviewers indicating that the optimization formulation and overall presentation are easy to follow.
The empirical evaluation is thorough and scalable, covering multiple model sizes, diverse architectures (including GQA), detailed ablation studies, and hardware throughput measurements, all showing consistent improvements over prior SVD-based methods.
The paper is well-written and accessible, effectively communicating the motivation, design choices, and empirical findings of dynamic rank allocation.


We thank all reviewers for their valuable comments and constructive feedback. We also  have addressed the weaknesses and concerns raised by reviewers in the authors’ rebuttal, where we provide additional clarifications, new analyses, and further justification for our design.

---

### Meta-Review · Area_Chair_1APY · 2026-01-07

**Summary:**

Initial scores are 6, 6, 2, 2. Reviewers acknowledged clear motivation, comprehensive experiments showing 15+ lower perplexity improvements and 5% higher reasoning accuracy. Concerns I consider most relevant for the decision are:
- limited novelty relative to RankDyna (EMNLP 2023 Findings) addressing similar dynamic rank problems,
- straightforward entropy-based metric without substantial algorithmic advance beyond SVD-LLM/Basis Sharing extensions,
- insufficient theoretical justification connecting effective rank to generalization or reconstruction fidelity, and
- missing comparisons with SVD-LLM v2 and unclear/potentially unfair experimental settings (FWSVD perplexities in thousands)

I've taken into account the authors' concerns about the potentially LLM-generated review by 51KQ with factual inaccuracies.

**Reviewer Concerns:**

**Addressed:**
- **RankDyna distinction (51KQ, NDjg):** Authors pointed out some differences: RankDyna targets fine-tuning efficiency using Taylor-expansion loss estimation during training; D-Rank targets post-training deployment using analytic spectral-entropy allocation without retraining.
- **Deployment feasibility (51KQ):** Authors clarified activation statistics used only once during offline compression (same as SVD-LLM/Basis Sharing), not during inference. No runtime overhead, fully compatible with prefix-cache frameworks (SGLang, vLLM, TensorRT-LLM).
- **Efficiency overhead (51KQ):** One-time preprocessing takes 18.9min for LLaMA-7B (identical to baselines). Inference faster than baselines: 2m58s vs. 3m13s (40% compression), 3m40s vs. 3m51s (30% compression) on WikiText-2.
- **Alternative metrics comparison (F3JZ, NDjg):** Authors tested stable rank (PPL 7.69) and 95% energy threshold (PPL 7.80) vs. effective rank (PPL 7.62) on LLaMA-7B. Stable rank fails to capture Q/K/V disparity (V:4.9-23.9, Q:1.3-2.0), insufficient to motivate balancing strategy.
- **Experimental settings clarification (NDjg):** Authors confirmed no fine-tuning/updates for D-Rank—pure post-training allocation. FWSVD perplexity (18,156) matches original SVD-LLM paper reproduction.
- **SVD-LLM v2 distinction (XMLc):** Authors explained SVD-LLM v2 uses heuristic loss-driven empirical balancing requiring calibration data; D-Rank uses analytic information-maximization with closed-form global allocation without activations/retraining. Orthogonal refinement techniques.
- **Ablation studies (XMLc):** Separate contributions quantified on LLaMA-7B C4 (20% compression): Basis Sharing 11.42 → dynamic rank 11.15 → balancing 11.03 → combined D-Rank 10.78.
- **Larger model validation (XMLc):** Added LLaMA-65B results: D-Rank 6.26 vs. Basis Sharing 6.42 PPL (40% compression).


**Outstanding:**
- **Novelty perception (51KQ, NDjg):** Despite clarifications on methodological differences from RankDyna and theoretical grounding, the rebuttal does not convincigly sustantiate the high level of novelty expected for this venue as the contribution can still be percieved as incremental extension of existing SVD-based methods rather than substantial algorithmic innovation.
- **SVD-LLM v2 direct comparison (XMLc):** Without released implementation or ablations isolating components, fair head-to-head comparison remains infeasible. Authors note orthogonality but cannot demonstrate empirical superiority.
- **Theoretical justification for effective rank (F3JZ, 51KQ, NDjg):** While authors cited information theory foundations [Roy & Vetterli 2007] and showed empirical superiority (effective rank PPL 7.62 vs. stable rank 7.69 vs. 95% energy 7.80), they did not convincingly explain why spectral entropy is fundamentally superior to alternatives (stable rank, numerical rank, nuclear norm) for compression. Citation of participation ratio concept does not justify why entropy-based metric optimally captures compression-relevant information diversity versus other spectral measures.

**Reviewer Scores:**

- **Reviewer F3JZ (initial: 6):** Would likely remain at 6. Positive initial assessment with concerns about novelty relative to RankDyna and attention-specific reallocation justification. Authors provided comprehensive RankDyna distinction and Q/K/V disparity analysis, but core novelty perception may persist. No explicit score update signal.

- **Reviewer 51KQ (initial: 2):** Would likely remain at 2, although there is a small chance they might increase to 4. Major concerns (deployment feasibility, efficiency overhead, system compatibility) are comprehensively addressed with clarifications on offline-only preprocessing, full framework compatibility, and faster inference times. However, fundamental perception of limited algorithmic advance beyond straightforward entropy-based allocation may remain.

- **Reviewer XMLc (initial: 6):** Would likely remain at 6 but may increase to 8. All specific requests addressed: SVD-LLM v2 distinction explained, ablation studies added (dynamic rank + balancing contributions isolated), larger model experiments conducted (LLaMA-65B). Positive initial assessment with constructive feedback fully incorporated.

- **Reviewer NDjg (initial: 2):** Will likely remain at 2. Critical concerns addressed but novelty perception and theoretical justification may still be unsatisfactory.

---

### Decision · Program_Chairs · 2026-01-26

Reject